# Comparative Growth and Economic Performances between Indigenous Swamp and Murrah Crossbred Buffaloes in Malaysia

**DOI:** 10.3390/ani11040957

**Published:** 2021-03-30

**Authors:** Amirul Faiz Mohd Azmi, Hasliza Abu Hassim, Norhariani Mohd Nor, Hafandi Ahmad, Goh Yong Meng, Punimin Abdullah, Md Zuki Abu Bakar, Jaizurah Vera, Nurain Syahida Mohd Deli, Annas Salleh, Mohd Zamri-Saad

**Affiliations:** 1Department of Veterinary Preclinical Sciences, Faculty of Veterinary Medicine, Universiti Putra Malaysia, Serdang 43400, Selangor, Malaysia; amirulfaizazmi@gmail.com (A.F.M.A.); norhariani@upm.edu.my (N.M.N.); hafandi@upm.edu.my (H.A.); ymgoh@upm.edu.my (G.Y.M.); zuki@upm.edu.my (M.Z.A.B.); jaizurahvera@gmail.com (J.V.); NurainSyahida@gmail.com (N.S.M.D.); 2Laboratory of Sustainable Animal Production and Biodiversity, Institute of Tropical Agriculture and Food Security, Universiti Putra Malaysia, Serdang 43400, Selangor, Malaysia; annas@upm.edu.my; 3Faculty of Sustainable Agriculture, Universiti Malaysia Sabah, Sandakan 90509, Sabah, Malaysia; puniminabdullah@gmail.com; 4Department of Veterinary Laboratory Diagnosis, Faculty of Veterinary Medicine, Universiti Putra Malaysia, Serdang 43400, Selangor, Malaysia; mzamri@upm.edu.my

**Keywords:** buffaloes, economy, growth, Murrah crossbred, Swamp

## Abstract

**Simple Summary:**

A buffalo breeding farm was selected to study the growth performance of Swamp and Murrah crossbred buffaloes. The farm was practicing extensive grazing system without supplementation since 2010 to 2011. In early 2012, the farm had implemented a new intervention to improve the growth performance via improving the feed and the feeding management. Farm records between 2010 to 2015 were analyzed for growth performance and partial budget analysis. So far, there is no comparative study done between Swamp and Murrah crossbred buffaloes in Malaysia. Therefore, in the present study, we aimed to study the differences in the biological and economical performances between Swamp and crossbred buffaloes in Malaysia. With a new intervention, a significant improvement was reported of the number of calves born, average birth weight, and reduced percentage of calf mortality rate, calving interval, and weaning age. Crossbred buffalo showed dominance in biological performance in terms of higher pre- and post-weaning daily weight gain and taking a shorter period to achieve market and breeding weight compared to Swamp buffaloes. Thus, reared Murrah crossbred buffaloes with new intervention management would give a farmer a higher profit return. However, with reared Swamp, the farmer potentially conserves the local indigenous breed of Swamp buffalo.

**Abstract:**

This study was conducted to compare the growth and economic performances between Swamp and Murrah crossbred buffaloes. The records of 108 Swamp and 276 Murrah crossbred buffaloes born between January 2010 and December 2015 were used in this study. The farm was practicing an extensive grazing system without supplementation from January 2010 to December 2011 (pre-intervention) and a new implementation of supplement in the feeding regime from January 2012 to December 2015 (post-intervention). The birth, weaning, and body weight at three monthly intervals, number of calves born, and mortality rate of calves at different years and during pre- and post-intervention were analyzed using a general linear model procedure. The interventions in 2012 had a positive effect on increasing the number of calves born for both breeds, average birth weight, economic performance, and reduce mortality calf rate. As a result, the birth weight of Murrah crossbred buffaloes was higher (36.63 ± 0.50 kg) than Swamp buffaloes (34.69 ± 0.40 kg) (*p* < 0.05). The average pre-weaning daily weight gain for Swamp and Murrah crossbred buffaloes was 0.73 and 0.98 kg/day (*p* < 0.05), while the average post-weaning daily weight gain was 0.39 and 0.44 kg/day, respectively (*p* < 0.05). The Swamp and Murrah crossbred buffaloes achieved the targeted market weight of 250 kg at 18 and 15 months old, respectively, while the targeted breeding weight of 385 kg was achieved at 30 and 26 months old, respectively. In this farm, on average a total of 64 calves were born yearly, with the ratio number of born calves per number of mated dams recorded higher in Murrah crossbred buffaloes as compared to Swamp buffalo (0.64 vs. 0.37) (*p* < 0.05). Furthermore, the average number of calves born in the post-intervention period (January 2012–December 2015) was significantly higher than in the pre-intervention period (January 2010–December 2011), respectively (Swamp: 23 vs. 8 and Murrah crossbred: 53 vs. 31, respectively) (*p* < 0.05). Partial budget method was used to estimate the net gain or loss between the two breeds. The average annual revenue was 2304.14 MYR (566.13 USD) for Swamp buffaloes and 4531.50 MYR (1113.39 USD) for Murrah crossbred buffaloes. The average annual cost saving was 340.02 MYR (83.54 USD) for Swamp and 215.75 MYR (53.01 USD) for Murrah crossbred buffaloes. On the other hand, annual added cost was 84.95 MYR (20.87 USD) for Swamp and 96.76 MYR (23.77 USD) for Murrah crossbred buffaloes. Therefore, the annual net benefit was 2559.21 MYR (628.80 USD) for Swamp and 4650.49 MYR (1142.63 USD) for Murrah crossbred buffaloes. As a conclusion, this study had shown that the higher average daily weight gain contributes to better cost savings, as shown by the crossbred buffaloes.

## 1. Introduction

There are two types of domestic water buffaloes or Asian buffaloes, the Swamp and river buffaloes and Murrah buffaloes [1] as shown in Figure 1 and Figure 2. The Swamp buffaloes are found mainly in Southeast Asia and China and are kept mainly for meat and draft power in paddy fields and oil palm plantation. However, Murrah buffaloes are found mainly in South Asia, especially India and Pakistan, and are kept mainly for milk. In the field, river and Swamp buffaloes can be differentiated by their morphology and behavior. Swamp buffaloes are ash or dark grey with a white chevron line on the neck either one or two stripes and have socks, while the tip of the tail and the horns are swept backwards [2,3]. They prefer to wallow in the marshland and mud and have large feet with slow steady movement that make them well suited for paddy land preparation in swampy waterlogged rice fields [4]. On the other hand, Murrah buffaloes have a black body with tightly and forwardly curled horns. They prefer to wallow in clean water [2]. Generally, Swamp buffaloes are smaller than the Murrah buffaloes, while the crossbred buffaloes have the same morphology as Murrah. However, they are smaller than Murrah, but bigger than the Swamp buffaloes.

In trying to improve the milk performance of buffaloes, crossbreeding between the Murrah and the Swamp buffaloes was attempted in many Southeast Asian countries, notably in the Philippines and Malaysia, with the hope that the resulted Murrah crossbred animals are bigger in size to produce better milk yield than the native Swamp buffaloes [5]. In Malaysia, Swamp buffaloes are raised purposely for meat and draught work, while Murrah crossbred buffaloes are raised for milk. Later, it was reported that milk yields from crossbred buffaloes are significantly higher than the local buffaloes [6]. However, the status of buffalo milk production in Malaysia is poorly understood and is currently under-developed. Milk production in dairy buffalo in Malaysia varies from 4.00 to 6.00 L/day [7] to approximately 8.00 L/day [8], which is still behind the potential of superior buffaloes at 15.00–20.00 L/day [9]. However, our local Murrah crossbred buffalo can only produce an average of 4.7 L of milk/day/animal per lactation period, producing on average a total of 1000 L of milk annually [9], and this is far from meeting the local milk demand. 

In general, buffalo is one of the ruminants with economic importance for small holder farmers in many developing countries in Asia [10]. Furthermore, due to farm record, they have a longer lactation period; hence, farmers potentially harvest more milk and generate more income. In fact, crossbred buffaloes mature earlier and have shorter calving interval, hence producing more calves in their lifetime [6]. Therefore, to estimate the potential improvement in the income from buffaloes, the economic performance between Swamp buffaloes and Murrah crossbred buffaloes should be compared. This will aid future farmers in choosing the beneficial buffalo breed.

According to Nanda et al. [11], reproductive and growth performance of buffaloes are generally poor. This was supported by previous study reports that buffalo management in a buffalo breeding farm in Sabah was practicing the traditional feeding management before 2012 and changed the feeding management in January 2012 [12]. Although there was a retrospective study, comparing the reproductive performance at that farm in 2004 until 2011, there was no comparative study on the differences in the biological and economical performances between Swamp and Murrah crossbred buffaloes in Sabah, Malaysia, during pre- and post-intervention implementation of new feeding regime. Therefore, the objective of this study was to analyze the growth and economic performances between the indigenous Swamp and Murrah crossbred buffaloes kept at a breeding farm in Sabah, Malaysia, and describe their performance following pre- (January 2010–December 2011) and post-interventions (January 2012–December 2015) with an improved rearing system.

## 2. Materials and Methods

### 2.1. Statement of Animal Rights

The study was performed and managed according to approval of Animal Utilization Protocol (AUP), Institutional Animal Care and Use Committee (IACUC), Universiti Putra Malaysia (Approval No. UPM/IACUC/AUP-R039/2012, on 2 February 2012). 

### 2.2. Study Area

This study was conducted at the Buffalo Breeding and Research Centre, Telupid, Sabah (5°30’ N, 117°7’ E). The farm consisted of 749 acres of land, and currently with a total of 405 heads of buffaloes. There were two types of buffalo breeds available in this farm, the Swamp and the Murrah × Swamp crossbred, representing 54% and 46% of the total buffalo population, respectively. The sample size in this study was represented by 10% to 15% of the total buffaloes in the farm, according to the Cochran method [13,14]. The sample size reflected proportions of the population in the farm, with the buffaloes selected randomly.

The crossbred animals were selected from the farms that were practicing natural breeding, which involved breeding between the pure Murrah males with Swamp females. The identification of these two subspecies of buffaloes was conducted based on conventional techniques (morphological or physiological characteristics). For genetic predisposition, dominance, and origin of crossbred, karyotypic (based on chromosomes number) and molecular identification techniques were used to identify the phylogeny of the animals or to compare the crossbred buffaloes with their purebred parents [15,16,17]. In addition, prior study on this farm had reported that phylogenetic tree and mtDNA analysis on Swamp buffaloes were genetically conserved and the crossbred buffaloes were dominantly Swamp according to the maternal lineage using d-loop mtDNA [17].

The 398.5 acres of pastureland were divided into two paddocks with establish pasture (*Brachiaria decumbens*). The Swamp and Murrah crossbred buffaloes in this farm were kept separated. Wallowing areas were available in each paddock and drinking water was available ad libitum. The buffaloes were monitored daily by farm supervisor on their health and nutritional management. The buffaloes were kept extensively and free to graze all day within the paddock enclosed by barbed wire. Even though the pasture in general was poorly maintained, the farm practiced an extensive one-month rotational grazing system without feed supplementation, and the intervals were determined according to the size of each paddock and through the supervision of the farm manager. This farm practiced natural breeding with male to female ratio of 1:20. Breeding season was between November and January each year, and pregnancy diagnosis was carried out every three months following breeding. As routinely practiced by the farm, upon reaching the market weight of 250 kg, the buffaloes were sold, mainly to the surrounding farmers for fattening. The farm practiced minimum breeding weight of 385 kg, and all buffaloes were weighed every three months. 

### 2.3. Pre-Intervention

At the start of the study, two years of farm records between January 2010 and December 2011 were selected and analyzed for growth performance parameters. At the same time, pasture samples were collected at six sites of 1 m^2^ area from the paddock for proximate analysis of the nutrient contents [18]. 

### 2.4. The Intervention

The intervention was implemented from January 2012 until December 2015 by improving the feeding management. This included the use of fertilizer (20 tons organic fertilizer with 500 kg of urea) and proper management to improve the pasture. Fertilizer was used on the pastureland twice a year. The animals were offered concentrate supplementation at the rate of 1.5 kg/animal/day (moisture: 9.64%, ash: 5.44%, crude fiber: 7.49%, crude fat: 5.46%, crude protein: 18.15%, gross energy: 15.74% MJ/kg). After six months, pasture and concentrate samples were collected from the paddocks and re-analyzed of the nutrient contents (Table 1). In January 2016, all data from 2010 to 2015 were gathered for further analysis.

### 2.5. Analysis of Growth Performance

A total of 108 Swamp (female: 60 and male: 48) and 276 Murrah crossbred (female: 152 and male: 124) buffaloes that were born between January 2010 and December 2015 were included in this study. The average number of female’s breeders also was recorded in this study since 2010 to 2015 (Swamp: 49 and Murrah crossbred: 72). The average number of calving rates per year, mortality rate, and average birth weight were recorded in this study. The number of calving rates per year was calculated by the average number of calves born per year divided with the number of mating female per year. All selected buffaloes had gone through the same management system described previously. Records of the selected animals in both sexes, which included the animal identification and breed, birth weight, weaning weight (at three months old), and the three-monthly body weights were further analyzed using statistical software. The average daily weight gain and the period taken to reach the targeted 250 kg market weight (taking on average 12 to 18 months) and the 385 kg breeding weight (taking on average 24 to 30 months) were calculated.

The average weight gain was determined as the weight difference between the final weight and initial weight divided by the number of days between the final and the initial weights. Similarly, records of the average body weight and differences in the three-monthly body weight of the breeder Swamp and Murrah crossbred buffaloes were analyzed. 

### 2.6. Statistical Analysis 

All data were collected and subjected using Microsoft Excel and were analyzed using software package SPSS (Statistical Package for the Social Science 25.0, Inc., Chicago, IL, USA). Comparisons between breeds, months, years, and pre- and post-intervention and their interaction for parameters body weight, average daily weight gains, calving rate per year, mortality rate of calves, and average birth weight of calves from 2010 until 2015, were performed using the general linear model (GLM) procedures, and the resulting P-values were corrected by Bonferroni’s method for multiple comparisons. Linear mixed effects models were utilized with both ‘Diets’ and ‘Breed’ as fixed effects to capture the appropriate structure for GLM, while the months, years, and pre- and post-intervention were considered as a random effect. For all the statistical tests used, results were considered significant at *p* ≤ 0.05.

### 2.7. Partial Budget Analysis

The economic analysis was done to compare the differences between pre- and post- intervention using partial budget where net gain or loss was estimated by subtracting total loss from the total gain [19]. The assumption of total gain included the sum of increased revenue and reduced costs, while total loss included the sum of decreased revenue and added costs. Increased revenue included the increased values in improved birth weight, increased number of calves, improvement number survival rate of calves, and sales calves per year. Decreased revenue included reduced values in shorter first calving age cost and shorter calving interval cost. Added costs/losses were the costs of fertilizer, flushing, cost of calf supplemental feed, cost of feed for increase total population of calf heifer, deworming drugs, and ID tag.

The biological input was collected from the farm records, which included the birth, death, weaning, and breeding records reviewed as the data for the study. Other biological inputs, including the average increase in the number of calves born, average increase in birth weight, average decrease in calf mortality, and average age at first calving, were collected from the farm record books and by interviewing the farm manager. The calving interval was calculated as the interval period between two successive calvings of each buffalo divided by the number of buffaloes in calving group, whereas the calving rate was recorded as the ratio number of born calves per number of mated dams. In this calculation, the first 12 weeks following calving were deducted from the calculation, assuming that 12 weeks were required for recovery in which the females were not used for breeding. The biological inputs collected are as in Table 2. Early age at first calving was assumed to be at 33 months old (assuming 24 months to reach 350 kg from birth was adding with nine months for successful pregnancy). 

For the economical inputs, the prices of live buffalo per kg and feed per kg, and the costs of feed per week per animal, the fertilizer, the deworming drugs, and the ID tag per animal were obtained from the farm records. Average value of sold animals per year was obtained from the buffalo sales record book. The cost of treatment involved only the deworming drugs. The economic input is as shown in Table 3.

There was no foregone revenue, as Murrah crossbred and Swamp breeds were assumed to be similar in aesthetic value. To complete the calculation for the items, in total 12 formulas (1–12) were derived followed the method by Moran [20] as below:

Increased Revenues:Improved birth weight = Estimation value of improved birth weight per year (MYR) = BW_mean_ × ∆R × $ _per kg calf_(1)
BW_mean_ = Improved mean calf body weight; ∆R = Change ratio number born calves per number of mated dams; $ _per kg calf_ = Price of calf per kg
Increased number of calves = Estimation value of increase number calf per year (MYR) = ∆*n*_improved number of calves born_ × ∆R × $ _per kg calf_(2)
∆*n*
_improved number of calves born_ = Improved number of calves born; ∆R = Change ratio number born calves per number of mated dams; $ _per kg calf_ = Price of calf per kg
Increased number of survival calves = Estimation number of survival calves per year (MYR) = ∆SR × ∆R × $ _per kg calf_(3)
∆SR = Increased in survival rate; ∆R = Change ratio number born calves per number of mated dams; $ _per kg calf_ = Price of calf per kg
Increased sales of buffaloes = Estimation number of calves sales (MYR) = AS_2012–2015_ − AS_2010–2011_(4)
AS_2012–2015_ = Average sales of buffaloes in 2012 to 2015; AS_2010–2011_ = Average sales of buffaloes in 2010 to 2011

Reduced cost:Shorter in first calving age (MYR) = Estimation of shorter calving interval after new feeding management cost (MYR) = ∆FCA × FC _week_(5)
∆FCA = Change in first calving age; FC _week_= Feed cost per week
Shorter in calving interval cost = Estimation of shorter first calving age after new feeding management cost (MYR) = (∆CI × FC _week_) + ($ _flushing_ × 2)(6)
∆CI = Change in calving interval; FC _week_ = Feed cost per week; $ _flushing_ = Cost of flushing per animal; 2 = number of flushing frequencies

Increased cost:Cost of organic fertilizer per year = Estimation of organic fertilizer per year (MYR) = $ _fertilizer per animal_(7)
$ _fertilizer per animal_ = Cost of fertilizer per animal
Flushing cost per year = Estimation of flushing cost per year (MYR) = *n*_flushing days_ × *n*_amount PKC (kg)_ × $ _PKC/kg_ × 3(8)
*n*
_flushing days_ = Number of flushing days; *n*
_amount PKC (kg)_ = Amount of PKC used (kg); $ _PKC/kg_ = PKC price per kg (MYR); 3 = number of PKC given to animals’ frequency
Calf feed cost (Supplemented feed) = Estimation of additional calf feed cost per year (MYR) = *n*_days (within 3months)_ × $ _feed animals/week_ × ∆R(9)
*n*
_days (within 3months)_ = considering supplement given once a week with duration of 3 month (12 days); $ _feed/animals/week_ = Cost of animal feed per week; ∆R = Change ratio number born calves per number of mated dams
Feed cost for increase in total calf-heifer population = Estimation of additional feed cost for total calf-heifer population per year (MYR) = ∆R × $ _feed/animal/year_(10)
∆R = Change ratio number born calves per number of mated dams; $ _feed/animal/year_ = Cost of feed calf-heifer per year
Cost for deworming = Estimation of additional in deworming cost per year (MYR) = ∆R × $ _deworming/animal_(11)
∆R = Change ratio number born calves per number of mated dams; $ _deworming/animal_ = Deworming cost per animal
Cost for ID tag = Estimation of additional cost for ID tag per year (MYR) = ∆R × $ _ID tag cost/animal_(12)
∆R = Change ratio number born calves per number of mated dams; $ _deworming/animal_ = ID tag cost per animal

Forgone Revenue:Assume similar quality present in both breed = The value given for both breeds were 0

The sum of the new revenue and the costs saved gave the value of total additional gains due to the intervention. The sum of revenue forgone and the new costs gave the value of total additional costs due to the intervention. The net benefit due to the intervention was the difference between total additional gains and total additional costs. The formula for the calculation is as below:Net benefit = (Increased revenue + Decreased cost) − (Revenue foregone + Increased costs)

## 3. Results

### 3.1. Growth Performance 

In general, the mean bodyweight patterns of Swamp and Murrah crossbred buffaloes from birth until 24 months old are shown in Table 2. There was significant (*p* < 0.05) difference in the birth weight between Swamp (34.69 ± 0.50 kg) and Murrah crossbred (36.63 ± 0.40 kg) buffaloes. Thereafter, the crossbred buffaloes showed significantly (*p* < 0.05) higher body weights at each of the three-monthly intervals. Interaction among breeds and three-month intervals were significantly (*p* < 0.05) different. In this farm, the weaning age was practiced at three months old. Thus, the body weight at three months old showed a significant higher in Murrah crossbred (124.7 ± 2.80 kg) as compared to Swamp (99.9 ± 3.10 kg) buffaloes. At 24 months old, Swamp showed significantly (*p* < 0.05) lower body weight of 320.70 ± 5.60 kg compared to 356.60 ± 5.90 kg for Murrah crossbred buffaloes (Table 4). The targeted 250 kg market weight was achieved at 18 months old for Swamp and at 15 months old for Murrah crossbred buffaloes (Figure 3).

### 3.2. Average Daily Weight Gain 

The mean (±SEM) average daily weight gain (ADG) for Swamp and Murrah crossbred buffaloes are summarized in Table 5. In this study, Murrah crossbred buffaloes showed significant (*p* < 0.05) higher in average daily gain compared to Swamp. Highest average daily weight gain was observed during the pre-weaning period, while the lowest was in the period between 12 and 24 months old. The pre-weaning ADG was 0.73 ± 0.03 kg and 0.98 ± 0.03 kg (*p* < 0.05) for Swamp and Murrah crossbred buffaloes, respectively. However, between 12 and 24 months old, the average daily weight gain did not differ significantly (*p* > 0.05) with 0.30 ± 0.01 and 0.29 ± 0.01 kg/day for Swamp and Murrah crossbred buffaloes, respectively. Moreover, when buffaloes reach more than 24 to 30 months old, the average daily weight gain by Murrah crossbred buffaloes (0.44 kg/day) was significantly (*p* < 0.05) higher compared with to the 0.39 kg/day for Swamp buffaloes. Thus, the Murrah crossbred buffaloes reached the targeted breeding weight of 385 kg by 26 months old, while the Swamp buffalo reached it by 30 months old. However, there was no significant record on the month intervals and interaction between months and breeds.

### 3.3. Number of Calves and Calf Quality 

In general, the post-intervention period showed significant (*p* < 0.05) improvement in number of calves born, average birth weight, and reduced mortality calf rate compared to the pre-intervention period. The number of calves born was taken from the farm record, where every single female produced calving annually, considering the period for pregnancy was nine months and 12 months of resting period. The numbers of calves born per year for Swamp and Murrah crossbred buffaloes are shown in Figure 4. In this farm, on average a total of 64 calves were born yearly with the ratio number born calves per number of mated dams recorded higher in Murrah crossbred buffaloes compared to Swamp buffalo (0.64 vs. 0.37) (*p* < 0.05). Over the study period, both Swamp and Murrah crossbred buffaloes showed an increasing trend of calves’ birth with an average annual increase of 18 calves for Swamp and 46 calves for the Murrah crossbred buffaloes; significantly (*p* < 0.05), calves’ birth was more than the Swamp buffaloes. During pre-intervention (2010–2011), the average number of calves born recorded significantly lower compared to post-intervention period (2012–2015) (Murrah crossbred: 31 vs. 53, Swamp: 8 vs. 23, respectively) (*p* < 0.05). Furthermore, the ratio number of calves born per number of mating females during pre-intervention (2010–2011) showed significant lower compared to post-intervention period (2012–2015) (Murrah crossbred: 0.52 vs. 0.63, Swamp: 0.40 vs. 0.45) (*p* < 0.05). However, there were no interaction (*p* > 0.05) showed between number of calves born with different years and pre- and post-interventions.

The improvement in calf quality was measured through the rate of calf mortality and the average birth weight. Figure 5 shows the rate of calf mortality in which the Swamp and Murrah crossbred buffaloes showed the average decreasing pattern of 7.4% and 8.1%, respectively, and significantly (*p* < 0.05) showed better improvement than the Swamp buffaloes. However, no interaction (*p* > 0.05) was recorded between calf mortality rate with different years and pre- and post-interventions. Other than calf mortality, improving birth weight was also considered as an increased in calf quality. Figure 6 shows the average birth weight of calves for Swamp and Murrah crossbred buffaloes. During the study period, Swamp buffaloes showed an average increment of 9.15 kg, which was significantly (*p* < 0.05) better than the 7.4 kg for the Murrah crossbred buffaloes. Furthermore, there were significant differences (*p* < 0.05) on average birth weight, years of birth, and interaction between average birth weight with years of birth and pre- and post-interventions. These data were included as increased revenue to the farm.

### 3.4. Calving Interval and First Calving Age

The average calving interval for Swamp buffaloes was 398 days (13 months) and 460 days (15 months) for the Murrah crossbred (*p* < 0.05). Both breeds showed the calving age at 33 months in 2015, reduction of 12.5 months for Swamp, and 6.5 months for Murrah crossbred buffaloes from 2010.

### 3.5. Partial Budget

Table 6 shows the increased revenue for Swamp and Murrah crossbred buffaloes. The increased revenue for both breeds came from the increased number of animals due to the increased number of calves being born and the improvements in mortality rate, the improved quality of the animals following improvement in birth weight, and the increased sales per year. The improved birth weight per animal of Swamp buffaloes was valued at 1.86 MYR (0.46 USD), while the Murrah crossbred buffaloes was 3.46 MYR (0.85 USD). The increased numbers of calves were valued at 4.13 MYR (1.02 USD) for Swamp and 12.10 MYR (2.97 USD) for Murrah crossbred buffaloes. The improvement number survival rate of Swamp and Murrah crossbred buffaloes were 2.02 MYR (0.50 USD) and 4.94 MYR (1.21 USD), respectively. The increased sales per year for Swamp and Murrah crossbred buffaloes were 2296.13 MYR (564.16 USD) and 4511.00 MYR (1108.35 USD), respectively. Therefore, the overall increase in revenue for Swamp buffaloes was 2304.14 MYR (566.13 USD), while the Murrah crossbred buffalo was 4531.50 MYR (1113.39 USD). The Murrah crossbred buffalo showed higher scale of increased revenue than the Swamp buffaloes.

Another input that was considered under gains was the decreased cost of production, which included the shorter calving interval cost and shorter first calving age cost. The shorter calving interval after new feeding management cost for Swamp buffaloes was valued at 90.00 MYR (22.11 USD), while for Murrah crossbred buffaloes it was at 46.80 MYR (11.50 USD). The shorter first calving age after the new feeding management cost was valued at 250.02 MYR (61.43 USD) for Swamp and 168.95 MYR (41.51 USD) for Murrah crossbred, respectively. Therefore, the total decreased cost for Swamp buffaloes was 340.02 MYR (83.54 USD) and for Murrah crossbred buffaloes was 215.75 MYR (53.01 USD). 

Total gains were calculated by the sum of total increased revenue and decreased cost. Therefore, total gain for Swamp buffalo was 2644.16 MYR (649.67 USD), while the Murrah crossbred buffalo was 4747.25 MYR (1166.40 USD), higher than the Swamp buffaloes.

In this study, the losses included the increased costs of production, which consisted of costs of fertilizer, flushing, supplemented feed, feed cost for increase total population of calf heifer, deworming, and ID tags. Table 6 also summarizes the increased costs for rearing Swamp and Murrah crossbred buffaloes. The cost of fertilizer for per animal Swamp and Murrah crossbred buffaloes was 64.79 MYR (15.92 USD). The cost of flushing the Swamp buffaloes was 13.26 MYR (3.26 USD), while the crossbred buffaloes was 16.80 MYR (4.13 USD). The cost of supplemented feed due to the increase in a number of calves was valued at 4.32 MYR (1.06 USD) for Swamp and 9.50 MYR (2.33 USD) for Murrah crossbred buffaloes. The increased total population calf heifer of Swamp buffaloes was valued at 2.45 MYR (0.60 USD), while the Murrah crossbred buffaloes was 5.39 MYR (1.32 USD). The costs of deworming and ID tags were 0.03 MYR (0.01 USD) and 0.10 MYR (0.03 USD), respectively, for Swamp, while the Murrah crossbred buffaloes were 0.10 MYR (0.03 USD) and 0.22 MYR (0.05 USD), respectively. In total, the additional cost for Swamp buffalo was 84.95 MYR (20.87 USD) and for the Murrah crossbred buffalo was 96.76 MYR (23.77 USD). The Murrah crossbred buffalo showed slightly higher additional cost.

### 3.6. Net Gains/Loss

The difference between total gain and total loss was the net change. The net change for Swamp buffalo was 2559.21 MYR (628.80 USD), while the Murrah crossbred buffalo was 4650.49 MYR (1142.63 USD).

## 4. Discussion

The birth weight of Murrah crossbred buffaloes, as shown earlier, was heavier than that of the Swamp buffaloes [5,21,22]. However, this study reported much heavier birth weights for both Swamp and Murrah crossbred buffaloes compared to the previous reports. In fact, the average of birth weights of Swamp buffaloes reported in Malaysia [23] and Thailand [24] were lower compared to the present study. Similarly, the average of birth weights of Murrah crossbred buffaloes reported in Sri Lanka [25] was also lower than the present study. In this study, the average of birth weight showed significant (*p* < 0.05) interaction with the year of birth. According to previous research, the birth weight is influenced by the season of birth, year of birth, sex of the calf, and parity of the dam [24,26,27]. However, it is believed that the better birth weights observed in this study might be due to the good feeding and husbandry managements [12,28]. Thus, this study was in agreement with our findings where the new feeding regime potentially improved average birth weight.

Following birth, both Swamp and Murrah crossbred buffaloes showed rapid pre-weaning growth, but the crossbred buffaloes showed significantly better rate, leading to a better overall daily weight gain of 0.86 kg/day compared to 0.65 and 0.74 kg/day at three months old reported by [29,30]. As a result, the average body weight of three-month-old buffaloes in the present study (112.29 kg) was higher than the 86.5 kg reported earlier [29]. A similar study revealed that approximately nine months were needed for the calves to achieve body weight of more than 100 kg, irrespective of the breed of the buffaloes [5]. 

In this study, it was observed that ADG at birth to weaning age (zero to three months) was higher than at four to 30 months old. Comparing the performance ADG of buffalo between 0–3 months up to 24 to 30 months, it is revealed that weight gains and growth rates were higher during the periods when the calves were offered milk than when offered solid feed. In addition, the higher ADG before weaning age could be due to underdevelopment of the rumen buffalo calves at an early age, as observed and reported by Abbas et al. [30].

In general, the Murrah crossbred buffaloes had significantly heavier bodyweight than Swamp buffaloes from birth until 24 months old, which is in line with buffaloes in Indonesia [31], Thailand [22], and the Philippines [5,21,32]. The body weights of the buffalo calves can be affected by many factors, such as the feeding management [33,34,35], breeds of buffalo [5,25,27], and environmental factors [24,36,37]. 

However, there were several factors contributing to the poor performance of animal growth at farm levels, namely improper nutritional management, climate change, seasonal stress, metabolic diseases, and mismanagement of farm [12,38]. Prior to intervention, nutritional analysis of the pasture before 2012 showed the average of crude protein, carbohydrates, ash, ether extract, and gross energy were low in value contents. The grass used by this farm had been the primary diet to boost performance of the animals. Additionally, another study has shown that *Brachiaria decumbens* were able to give the animals with nutritionally inferiority [39] when farms have proper pasture management. Thus, managing the farm with a proper management of pasture and supplementation diet in future will potentially make the buffalo have a better performance in growth, reproduction, and high production of quality milk to calves and industry over a long productive life [38,40]. Meanwhile, after the implementation of the intervention (2012–2015) via a new feeding regime by supplementing concentrate in the basal diet and improvement of pasture management, the nutritional contents of the buffalo’s diets were improved. 

Since the improvement of the diet due to post-intervention potentially improves the growth performance of both breeds, calves after weaning must be allowed to graze on improved pasture, be provided with a proper ratio of concentrate supplement to improve the growth performance, and be able to reach the breeding age in a shorter period [12,40]. This study also was in agreement with Vendramini et al. [41], where the study revealed a positive linear relationship between concentrate supplementation and ADG, liveweight gain, and stocking rate of early weaned calves of *Bos sp*. In addition, the new intervention revealed a better impact on the improvement of average birth weight, number of calving rate, and reduce calf mortality rate. Indeed, the number of calving rates in this study was in agreement with Wahid and Rosnina’s [9] findings, which indicated that the buffalo in Asia undergo year-round breeding and produces two calves every three years on average.

The reduction of calf mortality rate was shown before and after intervention from 26.83% to 19.81%, respectively. However, a study in Pakistan revealed an average of 17.98% mortality rate among buffalo calves [42], which was lower than this study. Other studies from India recorded that the mortality rate among dairy buffalo farms was 81.09% [43]. According to Panchasara et al. [42] and Othman et al. [12], the death of buffalo calves was usually reported during the third to fourth weeks of age, during certain seasons (monsoon, winter, or during heavy rainfall season) and improper feeding management, which had led to metabolic disease such as ketosis. 

Between 2011 and 2015, improved farm husbandry practices have resulted in an increased number of new-born calves and low rate of calf mortality, which had contributed to the increased revenue for both Swamp and Murrah crossbred buffaloes [12,28]. However, the Murrah crossbred buffaloes showed higher value due to the marked increase in the number of new-born calves and lower rate of calf mortality, making the population of crossbred calves much higher, which had brought more income to the farm. In addition, the Murrah crossbred buffaloes showed heavier birth weight; thus, they tended to reach weaning weight much earlier than the Swamp buffaloes. 

It was reported in this study that the reduction of calving intervals had occurred after implementing the intervention. The data on calving interval recorded in Murrah crossbred (13 months) and Swamp buffalo (15 months) were lower compared to what was reported in India, where Murrah buffaloes had an average of 15 months [44] and Swamp buffaloes had an average of 18 months [4].

Partial budgeting is an economic model providing information on consequences of certain change in farm procedure without any specific time pattern [45]. The data retrieved from partial budget allows the farmer to understand the cost of production changes that determine the profit margins, and it is critical to ensure the sustainable of farm industry [46]. In this study, early weaning is costlier for the farm due to the longer weaning to production period. This leads to a slightly higher additional cost of the Murrah crossbred buffalo. On the other hand, the birth weight could also affect the reproduction maturity of females [47], when the Murrah crossbred buffalo tends to reach the age at first calving earlier than the Swamp buffaloes. Thus, the females can reproduce earlier and for a longer period, which brings more economic benefit to the farm. Earlier age of first calving also reduces the cost of rearing heifers.

Although better birth weight was beneficial to the farm economically, it also brings extra cost, especially in terms of feed consumption. As Murrah crossbred buffaloes were born bigger, their feed consumption was also higher than the Swamp buffaloes. Thus, they need more feed to get enough energy and nutrition for maintenance and growth, and this adds cost to the farm. Nevertheless, the Murrah crossbred buffaloes produced significantly higher overall net gains than the Swamp buffaloes. Therefore, farmers can reduce the rearing cost and earn more profit by rearing Murrah crossbred buffaloes and selling them at an earlier age either for slaughter or fattening purposes. Estimating costs via partial budget for rearing young buffalo could improve the farmer’s awareness and improve decision making in either rearing Murrah crossbred or Swamp buffalo, which could increase economic sustainability, meat, and milk production, as well as conserve the local indigenous breed in Malaysia.

## 5. Conclusions

In conclusion, the use of supplementation is a good practice with favorable findings when the system of buffalo rearing is addressed for meat and milk purposes. The early weaning of the buffalo calves, calving interval, reducing calf mortality rate, and improved calf birth weight may be a successful practice when having feed of good nutritional quality and proper pasture management. In future, this allows the calves to have early development of rumen physiology and metabolic function. Murrah crossbred buffaloes showed significantly heavier birth weight and better daily weight gains and thus reached market and breeding weights at a significantly younger age as compared to Swamp buffaloes. Subsequently, they resulted in better income gain for the farm when rearing Murrah crossbred buffaloes (4650.49 MYR) as compared to Swamp buffaloes (2559.21 MYR), whose crossbred buffaloes contribute to better cost savings and high net gains. The use of new intervention practice toward buffalo calves according to the rearing of both breeds has given a better extent of efficiency in the growth performance of the stock.

## Figures and Tables

**Figure 1 animals-11-00957-f001:**
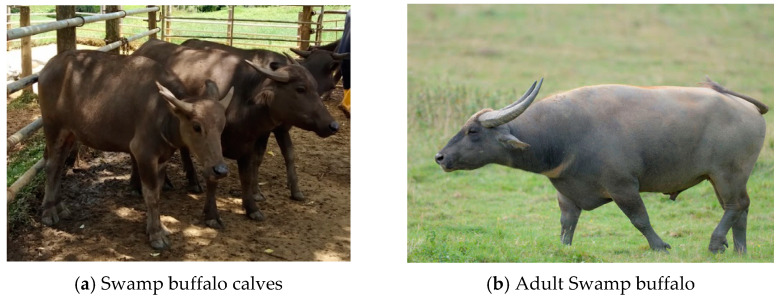
The Swamp buffaloes; (**a**): Swamp buffalo calves; (**b**): Adult Swamp buffalo.

**Figure 2 animals-11-00957-f002:**
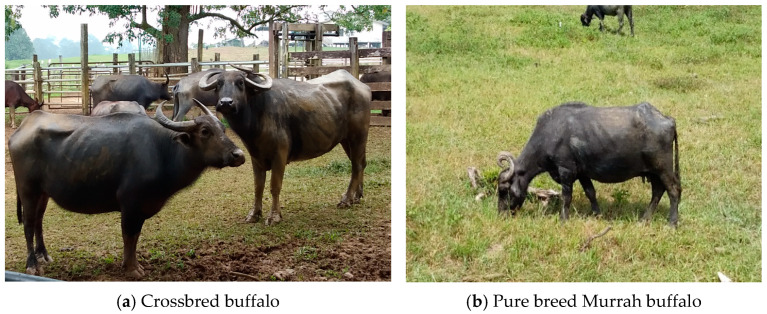
The crossbred and pure breed Murrah buffaloes; (**a**): Crossbred buffalo, (**b**): Pure breed Murrah buffalo.

**Figure 3 animals-11-00957-f003:**
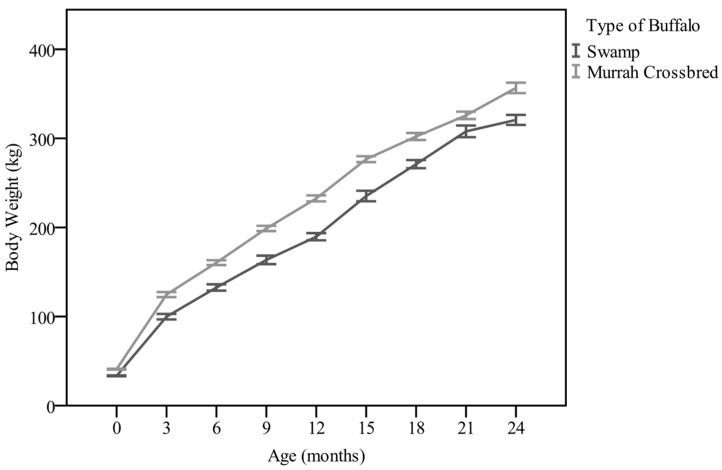
The body weight patterns of Swamp and Murrah crossbred buffaloes for both sexes.

**Figure 4 animals-11-00957-f004:**
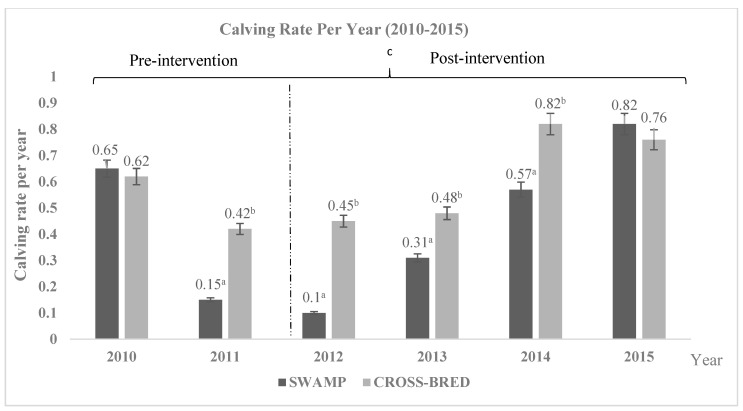
Total number of calving rates per year of Swamp and Murrah crossbred buffaloes between 2010 to 2015. There are increasing patterns for both breeds, and the crossbred buffaloes show significantly (*p* < 0.05) higher number than the Swamp buffaloes. ^a,b^ Different superscripts indicate significant difference of calving rate at *p* < 0.05; ^c^ indicating significant (*p* < 0.05) difference comparing between pre- and post-intervention. There is no significantly different (*p* > 0.05) interaction between number of calves born with different years of birth and pre- and post-intervention.

**Figure 5 animals-11-00957-f005:**
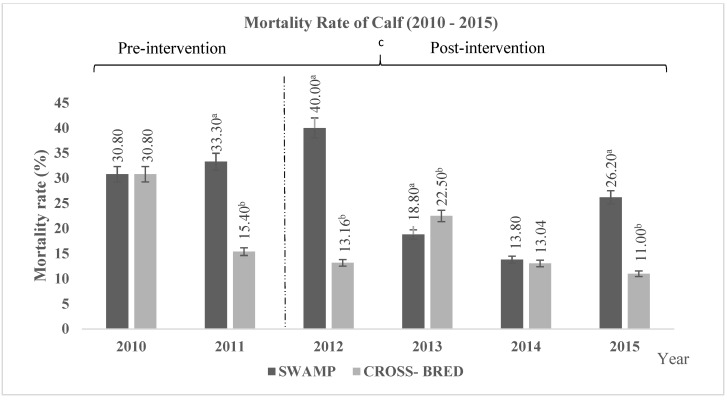
The rate of calf mortality among Swamp and Murrah crossbred buffaloes between 2010 to 2015. Based on Figure 5, the Murrah crossbred buffaloes show a significantly (*p* < 0.05) better improvement in the rate of calf mortality than the Swamp buffaloes. ^a,b^ Different superscripts indicate significant difference of mortality rate at *p* < 0.05; ^c^ indicating significant (*p* < 0.05) difference comparing between pre- and post-intervention. There is no significantly different (*p* > 0.05) interaction between number of calves born with different years of birth and pre- and post-intervention.

**Figure 6 animals-11-00957-f006:**
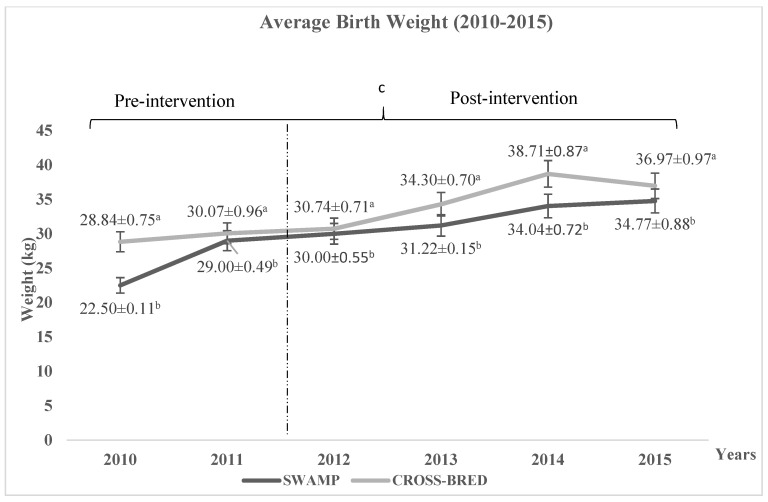
Average birth weight pattern of the Swamp and Murrah crossbred buffaloes between 2010 to 2015. Based on the Figure 6, the Swamp shows significantly (*p* < 0.05) better improvement than the Murrah crossbred buffaloes. ^a,b^ Different superscripts within the same year indicate significant difference of birth weight at *p* < 0.05; ^c^ indicating significant (*p* < 0.05) difference comparing pre- and post-intervention. There is a significantly different (*p* < 0.05) interaction between average birth weight with different years of birth and pre- and post-intervention.

**Table 1 animals-11-00957-t001:** Nutritional composition of diet on pre-intervention (January 2010–December 2011) and post-intervention (January 2012–December 2015).

Nutrient Composition	Pre-Intervention (*Brachiaria decumbens*)	Post-Intervention (*Brachiaria decumbens* + Concentrate)
^1^ DM (%)	99.50	99.49
Ash (% DM)	5.09	5.69
Crude fiber (% DM)	26.03	23.73
Ether extract (% DM)	2.03	2.92
Crude protein (% DM)	6.09	8.08
^2^ NDF (% DM)	64.27	57.96
^3^ ADF (% DM)	33.86	28.70
^4^ ADL (% DM)	3.55	3.32
Carbohydrate (% DM)	59.43	61.53
Gross energy (MJ/kg)	11.07	12.1
Hemicellulose (% DM)	30.41	29.25
Cellulose (% DM)	30.32	25.38

The data are the mean of triplicate analyses of each diet.^1^ DM: dry matter; ^2^ NDF: neutral detergent fiber, ^3^ ADF: acid detergent fiber, ^4^ ADL: acid detergent lignin.

**Table 2 animals-11-00957-t002:** Biological parameters used in the partial budget analysis.

Variables	Inputs	
	Swamp Buffalo	Murrah Crossbred
Difference ratio number of calves per number of mated dams between post- and pre- intervention	0.05	0.11
Difference in number of calves born between post- and pre-intervention (head)	15	20
Difference in calves birth weight between post- and pre-intervention (kg)	6.76	5.72
Difference in calve survival rate between post- and pre-intervention (%) ^1^	7.35	8.17
Difference in first calving age between post- and pre-intervention (months)	12.5	6.5
Difference in calving interval between post- and pre-intervention (months)	33.17	21.91
Average sales of buffalo calves (2010–2011)	8647.25	26,247
Average sales of buffalo calves (2012–2015)	10,943.38	30,758

All the calculation had been done between post- (2012–2015) and pre-intervention (2010–2011) period. ^1^ Survival rate: 100—mortality rate.

**Table 3 animals-11-00957-t003:** Economic parameters used in the partial budget analysis.

Variables	Price (MYR) ^1^
Buffalo calves per kg ^2^	5.50
Price of PKC per kg	0.80
Feed cost per week (MYR/animal)	7.20
Feed cost per year calf heifer (MYR/animal)	49.00
Fertilizer cost (MYR/animal)	64.79
Deworming cost (MYR/animal)	0.50
ID tag cost (MYR/animal)	2.00
Flushing cost (MYR/animal)	5.60

^1^ All amounts are in Malaysian Ringgit (MYR); ^2^ The price of live buffalo based on average market price since 2010 to 2015.

**Table 4 animals-11-00957-t004:** The body weight of Swamp and Murrah crossbred buffaloes for both sexes at three-month intervals (Mean ± SEM).

	Body Weight (kg)								
Month	Birth	3 *	6	9	12	15	18	21	24
Swamp (*n* = 35)	34.7 ± 0.50 ^a^	99.9 ± 3.10 ^bc^	132.6±3.50 ^c^	163.6 ± 4.80 ^d^	189.8 ± 4.00 ^e^	235.3 ± 5.90 ^f^	271.1 ± 4.60 ^g^	307.9 ± 6.50 ^h^	320.7 ± 5.60 ^i^
Murrah Crossbred (*n* = 35)	36.6 ± 0.40 ^b^	124.7 ± 2.80 ^bc^	160.5±2.70 ^d^	198.8 ± 3.00 ^e^	232.6 ± 3.40 ^f^	276.6 ± 3.30 ^g^	302.1 ± 3.90 ^h^	325.8 ± 4.20 ^i^	356.6 ± 5.90 ^j^
*p*-Value									
Breeds	<0.05								
Months	<0.05								
Interactions	<0.05								

^a,b,c,d,e,f,g,h,i,j^ Different superscripts within the same column indicate significant difference *p* < 0.05, * calves weaned at 3 months old.

**Table 5 animals-11-00957-t005:** Average daily weight gain (kg) of Swamp and Murrah crossbred buffaloes for both sexes (Mean ± SEM).

Month	0–3 *	3–12	12–24	24–30
Swamp (*n* = 35)	0.73 ± 0.03 ^a^	0.32 ± 0.02 ^b^	0.30 ± 0.01 ^b^	0.39 ± 0.07 ^b^
Murrah crossbred (*n* = 35)	0.98 ± 0.03 ^c^	0.37 ± 0.01 ^bd^	0.29 ± 0.01 ^be^	0.44 ± 0.08 ^bf^
*p*-Value				
Breeds	<0.001			
Months	0.304			
Interactions	0.108			

^a,b,c,d,e,f^ Different superscripts within the same column indicate significant difference *p* < 0.05, * calves weaned at three months old.

**Table 6 animals-11-00957-t006:** Partial budget analysis on intervention of feed supplementation for Swamp and Murrah crossbred (per female breeder per year).

Increases in Net Income			Decreases in Net Income		
Breed	Swamp	Murrah Crossbred	Breed	Swamp	Murrah Crossbred
Added Income Due to Change	MYR	MYR	Added Cost Due to Change	MYR	MYR
^1^ Improved birth weight	1.86	3.46	^7^ Fertilizer	64.79	64.79
^2^ Increased number of calves	4.13	12.10	^8^ Flushing	13.26	16.80
^3^ Improvement number survival rate of calves	2.02	4.94	^9^ Calf supplemental feed cost	4.32	9.50
^4^ Increased sales of calves per year	2296.13	4,511.00	^10^ Feed cost for increase total population of calf -heifer	2.45	5.39
			^11^ Deworming	0.03	0.06
			^12^ ID tag	0.10	0.22
(A) Total increase revenue	2304.14	4,531.50	(B) Total increased cost	84.95	96.76
Reduced cost due to change			Reduced income due to change		
^5^ Shorter calving interval cost	90.00	46.80	Assume similar quality	0	0
^6^ Shorter first calving age cost	250.02	168.95			
(C) Total decrease cost	340.02	215.75	(D) Total Foregone Revenue	0	0
E. Subtotal added gains (A + C)	2644.16	4,747.25	F. Subtotal added costs (B + D)	84.95	96.76
^13^ New Benefit (E − F)	Swamp = MYR 2644.16 − MYR 84.95 = MYR 2559.21 (USD 628.80)				
	Murrah cross = MYR 4747.25 − MYR 96.76 = MYR 4650.49 (USD 1142.63)				

^1^ USD = 4.07 MYR Currency Conversion 5 March 2021, MYR Malaysian Ringgit. The status quo of the partial budget analysis of the buffalo farm management is an extensive system. The change of the farm management system is the feed supplementation for Swamp and Murrah crossbred. The change that occurs reflects a semi-intensive farm management system. The additional cost to the farm due to change in farm management (pre- to post-intervention) increased. In return, the farm has gained their income due to improvement of birth weight, survival rate of calves, and increased number of calves and sales of calves per year. The reduced cost due to change of management also has increased when shorter calving interval period and first calving age are taken after new intervention. ^1–13^: ^1^ Estimation value of improved birth weight per year (MYR); ^2^ Estimation value of increase number calf per year (MYR); ^3^ Estimation value of selling calves to other farms for fattening per year (MYR); ^4^ Estimation number of calves sales; ^5^ Estimation of shorter calving interval after new feeding management cost (MYR); ^6^ Estimation of shorter first calving age after new feeding management cost (RM); ^7^ Estimation of organic fertilizer per year (RM); ^8^ Estimation of flushing cost per year (MYR); ^9^ Estimation of additional calf feed cost per year (MYR); ^10^ Estimation of additional feed cost of total population of calf-heifer per year (MYR); ^11^ Estimation of additional in deworming cost per year (MYR); ^12^ Estimation of additional cost for ID tag per year (MYR); ^13^ Estimation of net gain or loss per year (MYR).

## Data Availability

The datasets used and/or analyzed during the current study are available from the corresponding author on reasonable request.

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
