# Peer review of "Comparative Growth and Economic Performances between Indigenous Swamp and Murrah Crossbred Buffaloes in Malaysia"

_animals, 2021, doi:10.3390/ani11040957_

Round 1

Reviewer 1 Report

The authors have modulated deeply the manuscript. However, there are still some big incongruences that make the study difficult to understand, especially in the economic analysis. Moreover, the statistical analysis applied to analyze the effect of the intervention and the results obtained are not clear.

Some minor details (there are more but aI just specify two of them): three different statistical analyses (a*, b* and c*) but is difficult to understand which one is developed. The citations added in the manuscript do not follow the norms used in the rest of the manuscript.

Author Response

Thank you.

Reviewer 2 Report

The new manuscript is really changed to the better from animals-1032583-peer-review impressing. Congratulation. For me, there are only minor tasks to be performed before the manuscript can be published.

Line

Comment

55

Always space between USD and count (whole text); Give also here the date of exchange calculation.

88

animal? Check sentence.

92

The white or plain text has numbers for literature, but the new yellow text here has name, year for literature. Change everything to number and check literature. Whole text.

93

what does it mean: mmetric? This is not a SI value.

178

space between 2015 and were

446

not red

498

weight

529

from

581

States

603

Here you have to write something

Author Response

Thank you.

This manuscript is a resubmission of an earlier submission. The following is a list of the peer review reports and author responses from that submission.

Round 1

Reviewer 1 Report

The study is well conducted and the paper is in general written in a very clear way. However, there are some main issues related to the estimation of costs and statistical analyses that should be reviewed by the authors.

As a general main observation:

Please note that the abstract indicates “This study was conducted to compare the growth and economic performances between Swamp and Murrah crossbred buffaloes”. The first objective (growth) is correctly analysed but the development of the second objective (economic performance) is confuse. The paper is addressed in a way that seems to analyse the differences in income in that farm due to improvement of management along time and to the use of two different breeds. However, the objective is comparing the economic performance between both breeds, thus it has to be addressed analysing differences in profitability for a farmer depending on the breed used and expressed by female, or female and year, specially if you have different number of animals in each breed. In my opinion, improvement of management, birth weight or mortality do not have to be included, as the profitability is studied at a given moment and not along time. The analysis would be, moreover, comparably to other situations than this farm. Please, in case the analysis was developed as indicated as I suggest, clarify it in the text in another way in the text in order to avoid misunderstandings.

In general:

Please, indicate the origin of the crossbreed animals. Are they obtained by from crossing pure males and females from both breeds (thus, there animals analysed are all what in genetics is considered F1’s) or did you created a crossed population by initially crossing both breeds? This is an important point since in the first case, the higher performance could be due to both having half percent of the genetics from Murrah but also due to the heterosis, which is not shown in further generations. On the other hand, if Murrah has an economic advantage, why whould farmers rear the crossbreed animal instead of the pure breed Murrah?

Some specific points:

Weaning age and weaning weight are mentioned in M&M but they are not further shown or analysed.

Please specify the number of animals per sex.

Point 2.2. Please, indicate how you can calculate parameters between 250 and 385 kg if animals were sold to farmers.

Lines 92 to 95: “The calving interval was calculated as interval period between two successive calving of each cow after the first 12 weeks following calving were deducted from the calculation, assuming that 12 weeks were required for recovery and the affected females were not used for breeding. Early age at first calving was assumed to be at 33 months old”. This sentence is not clear. Please review it. Moreover, could you please specify what you mean with "affected females"?

Point 2.4.: Please indicate the statistical model used for each variable. Is the sex considered as a fixed effect?

Tables 1 and 2: “Different superscripts within the same row indicate significant difference” Please review if you mean column instead of row.

Table 2. There is a column indicating the weight gain for animals with more than 24 months of age. Is this variable commented in the M&M? Please specified wherever it corresponds: From 24 to which age? Are pregnant females included in this group? Why weight gain increases with respect to the gain observed in younger ages?

Line 146: It seems that mortality and weight at birth was improved in both breeds. As commented in the first comments, this improvement might be due to improvements in management, feeding, maybe environment effects, etc. However, it is not clear why this has to be considered in the partial budget, as is not related to differences in profitability when if using different lines. Please explain why you include this improvement observed with time.

Figure 2: The number of calves born was recorded, as you mention in M&M. However, the analysis of this variable is not commented in the abstract and in the statistical analysis, so I suggest adding this information.

Moreover, according to the figure, it seems that you compare statistically the number of calves, but the number of females giving birth varied between breeds, thus information obtained from the analysis seems confusing and not conclusive. Please analyse, instead, the number of calves born per delivery.

Figure 4: “Different superscripts indicate significantly difference at P < 0.05” I suggest “Different superscripts within the same year indicate significantly difference at P < 0.05”. On the other hand, please check that the Murrah breed is also shown in the legend, as it is not shown in my version.

Line 182: The is an “(“ left before USD.

Table 3: The table 3 is difficult to understand:

In the title, do you mean "increased revenue" or "added revenue"? In case both terms are the same, please use only one of the terms along the whole paper.

Why there are very few animals in the analysis?

Is this expressed per female?

As I mentioned before, it is difficult to understand why the improvement in birth weight or the reduction of calving interval or cost of heifer observed along time might influence the income of a farmer in a given moment of time. I find that the same occurs with the terms "reduced calving interval" and "reduced cost of heifer".

At the same of time, please specify what you mean with "number of calves" and "sales per year". Units are different and both terms cannot be added to obtain a total. Do you maybe mean "income from selling calves to other farms for fattening" and "income from selling calves to the slaughterhouse"?

Table 4: Please indicate what you mean with “Total population”.

Author Response

Dear Reviewer,

Thank you for giving me the opportunity to submit a revised draft of my manuscript titled “Comparative growth and economic performance between indigenous Swamp and Murrah crossbred buffaloes in Malaysia” which submitting for special consideration of publication as an original research article in your reputable journal, Journal of Animals. The authors appreciate the time and effort that the reviewer has dedicated to providing your valuable feedback on my manuscript. The authors are grateful to the reviewer for their insightful comments on my paper. The authors have been able to incorporate changes to reflects most of the suggestions provided by the reviewer. The authors have highlighted the changes within the manuscript.

As attached together here is a response to the reviewer comment and concern. In addition to the above comments, all spelling and grammatical errors pointed out by the reviewer have been corrected. Please see the attachment.

Thank you for your consideration of this manuscript.

Reviewer 2 Report

see attachment

Author Response

(The authors gave the same response as above.)

Reviewer 3 Report

Authors have contributed an important aspect of Buffalo rearing economics. A brief description of both breeds, along with its phenotypic characters may be added in the introduction part, as well as milk yield performance.

It is advised to incorporate a representative picture of both buffalo types, especially of Murrah crossbred.

Another important aspect that should be inserted is about the feed and feeding management of these types during the study period. For example, type of grazing pasture, their nutritive values besides the description of concentrate offered.

Author Response

(The authors gave the same response as above.)

Reviewer 4 Report

Dear authors

the paper was approximately structured. The results shown in the abstract was either wrong or there were no significant differences, ("The average birth weight of crossbred buffaloes was 36.63+/-5.18 kg, significantly (P<0.05) higher than the 34.69+/-5.28 kg birth weight of swamp buffaloes"). In the abstract, there was no conclusions, but there was comments, "This was due to the higher average daily weight gain and better cost savings shown by the crossbred buffaloes".

The introduction does not correctly present the status of current research in this area, and the scope of this work is not explained.

The experimental design is not correct, and the statistical analysis must be revised.

For these reasons I indicate the rejection of this article

Best regards.

Author Response

(The authors gave the same response as above.)
